# The RNA-Binding Ubiquitin Ligase MEX3A Affects Glioblastoma Tumorigenesis by Inducing Ubiquitylation and Degradation of RIG-I

**DOI:** 10.3390/cancers12020321

**Published:** 2020-01-30

**Authors:** Francesca Bufalieri, Miriam Caimano, Ludovica Lospinoso Severini, Irene Basili, Francesco Paglia, Luigi Sampirisi, Elena Loricchio, Marialaura Petroni, Gianluca Canettieri, Antonio Santoro, Luca D’Angelo, Paola Infante, Lucia Di Marcotullio

**Affiliations:** 1Department of Molecular Medicine, Sapienza University, Viale Regina Elena 291, 00161 Rome, Italy; francesca.bufalieri@uniroma1.it (F.B.); miriam.caimano@uniroma1.it (M.C.); ludovica.lospinososeverini@uniroma1.it (L.L.S.); irenebasili.ib@gmail.com (I.B.); marialaura.petroni@uniroma1.it (M.P.); gianluca.canettieri@uniroma1.it (G.C.); 2Dipartimento di Neurologia e Psichiatria, Neurochirurgia, Sapienza University, Viale dell’Università 30, 00185 Rome, Italy; francesco.paglia@hotmail.it (F.P.); luigi.sampirisi@gmail.com (L.S.); antonio.santoro@uniroma1.it (A.S.); Lu.Dangelo@policlinicoumberto1.it (L.D.); 3Center for Life Nano Science (CLNS@Sapienza), Istituto Italiano di Tecnologia, Viale Regina Elena 291, 00161 Rome, Italy; 4Laboratory Affiliated to Istituto Pasteur Italia-Fondazione Cenci Bolognetti-Department of Molecular Medicine, Sapienza University, Viale Regina Elena 291, 00161 Rome, Italy

**Keywords:** glioblastoma multiforme (GB), ubiquitylation, E3-ligase, MEX3A, RIG-I

## Abstract

Glioblastoma multiforme (GB) is the most malignant primary brain tumor in humans, with an overall survival of approximatively 15 months. The molecular heterogeneity of GB, as well as its rapid progression, invasiveness and the occurrence of drug-resistant cancer stem cells, limits the efficacy of the current treatments. In order to develop an innovative therapeutic strategy, it is mandatory to identify and characterize new molecular players responsible for the GB malignant phenotype. In this study, the RNA-binding ubiquitin ligase MEX3A was selected from a gene expression analysis performed on publicly available datasets, to assess its biological and still-unknown activity in GB tumorigenesis. We find that MEX3A is strongly up-regulated in GB specimens, and this correlates with very low protein levels of RIG-I, a tumor suppressor involved in differentiation, apoptosis and innate immune response. We demonstrate that MEX3A binds RIG-I and induces its ubiquitylation and proteasome-dependent degradation. Further, the genetic depletion of MEX3A leads to an increase of RIG-I protein levels and results in the suppression of GB cell growth. Our findings unveil a novel molecular mechanism involved in GB tumorigenesis and suggest MEX3A and RIG-I as promising therapeutic targets in GB.

## 1. Introduction

Glioblastoma multiforme (GB) is the most frequent and malignant primary tumor of the central nervous system [1], with a median overall survival of about 15 months [2,3,4]. GB is particularly difficult to treat, given its intense vascularization, rapid progression and high resistance to standard treatments [2]. The current therapeutic protocol for GB patients consists of surgical resection of tumor mass and subsequent concomitant radiotherapy and chemotherapy. However, these approaches show very limited effectiveness, resulting in a high rate of relapse and subsequent deterioration of the patient’s neurological and physiological status [2,5].

In the recent years, several genetic and epigenetic aberrations in molecular pathways (i.e., WNT and Hedgehog signaling) [6,7,8,9] have been associated with GB onset and progression, representing potential therapeutic targets and biomarkers for early prognosis [2,3,5]. Hence, the identification and characterization of new molecular players involved in GB tumorigenesis is essential for developing more effective and innovative therapies against this aggressive malignancy.

Ubiquitylation is a post-translational modification that controls a wide range of cellular functions (i.e., protein degradation, endocytosis and trafficking) and the most important physiological processes [10,11]. Ubiquitylation is mediated by an enzymatic cascade, in which the E3-ubiquitin ligases are the main players, responsible for the recognition of specific substrates and the final transferring of ubiquitin moieties onto target proteins.

Deregulation or mutations of E3-ubiquitin ligases have been associated with several human tumors; for this reason, they are considered to be promising targets for novel anticancer therapies [12,13,14]. At present, very little information is available about the role of E3 ligases and ubiquitylation processes in GB development and progression [15,16,17].

In this regard, we evaluated the expression levels of catalytic E3-ubiquitin ligase complex components [18] and F-box proteins of the SCF E3 ligase families in GB. From a preliminary gene expression analysis using publicly available datasets, we found a marked overexpression of MEX3A, and we investigated its potential involvement in GB biology. 

MEX3A belongs to an evolutionary conserved RNA-binding ubiquitin ligase protein family composed of four members (MEX3A-D). These proteins are the homologs of *Caenorhabditis elegans* MEX3, a translational repressor involved in germline totipotency and in the specification of posterior blastomere identity during embryogenesis [19]. Human MEX3 proteins consist of two N-terminal K homology (KH) domains, which provide RNA-binding ability, and a C-terminal RING finger module that confers ubiquitin E3-ligase activity [19]. MEX3 proteins play pivotal role in self-renewal and differentiation processes, with implications regarding stemness and carcinogenesis [20,21,22,23,24,25]. In particular, alterations of MEX3A activity have been described in gastric, colorectal and bladder cancers, although its mechanism of action and its potential substrates remain still elusive [21,22,23,24,26]. To date, only the RING finger domain of human MEX3C has been functionally and structurally characterized, and Retinoic acid-inducible gene I (RIG-I) is the unique substrate described for this E3 ligase [27]. Interestingly, the ubiquitylation of RIG-I mediated by MEX3C leads to RIG-I activation, without interfering with its degradation and protein stability [28].

RIG-I is a critical cytosolic pattern recognition receptor (PRR) that acts as RNA sensor to activate innate antiviral immunity and interferon (IFN) production [29]. However, recent findings highlighted the role of this protein in other cellular functions, as well as therapy resistance and expansion of cancer cells [30,31]. Furthermore, RIG-I works as a tumor suppressor in several tumor types, including GB [32,33].

In this study, we observed that MEX3A is up-regulated in GB and correlates with low RIG-I protein levels. We demonstrated that MEX3A interacts with RIG-I and impairs its protein stability by inducing its ubiquitylation and proteasome-dependent degradation. Interestingly, the genetic depletion of MEX3A results in the impairment of GB cell proliferation, providing new insights on GB tumor biology and identification of potential and innovative therapeutic approaches.

## 2. Results

### 2.1. MEX3A Expression is Up-Regulated in GB

To examine the expression of catalytic E3 ubiquitin ligases [18] and F-box proteins in GB, we first analyzed their expression in Gene Expression Profiling Interactive Analysis (GEPIA) (http://gepia.cancer-pku.cn/) (Figure 1A). We focused our attention on E3 ligases significantly up-regulated and whose molecular functions in GB tumorigenesis have not yet been described. Among them, we found the mRNA expression levels of the RNA-binding ubiquitin E3-ligase *MEX3A* strongly up-regulated in GB specimens compared to normal brain tissues (Figure 1B and Appendix A). This evidence was further confirmed in smaller datasets available at the R2 Genomics and Visualization Platform (http://r2.amc.nl) (Appendix A). These data were then validated in 27 GB specimens (Table 1) by assessing the mRNA expression levels of *MEX3A*. As shown in Figure 2A, we confirmed that the expression of *MEX3A* was significantly higher in GB samples compared to paratumor tissues, used as control.

### 2.2. MEX3A Up-Regulation Correlates with Low Protein Levels of RIG-I in GB

At present, ubiquitylation substrates of MEX3A have not been identified. We first hypothesized that this E3-ubiquitin ligase could be involved in the regulation of RIG-I, a tumor suppressor that controls differentiation, apoptosis and tumorigenesis processes. This idea was supported by the evidence that RIG-I stability and activity is modulated by the E3-ligases RNF122 [34] and MEX3C (the MEX3A paralogue) [20], whose mRNA expression levels were found to be up-regulated in GB (Appendix A and Figure 2C) [20,28,34].

Given the double function of MEX3A as RNA-binding protein and E3-ubiquitin ligase [19], we first analyzed the mRNA expression levels of *RIG-I* in GB. As shown in Figure 3A, we did not observe significant modulation of mRNA levels of *RIG-I* in GB specimens compared to noncancerous brain tissues.

Conversely, an opposite trend of MEX3A and RIG-I protein levels was observed in GB samples. Indeed, RIG-I protein levels were strongly decreased in GB tissues, in which high MEX3A protein levels were found; an inverse trend of the protein expression was observed in the corresponding paratumor samples (Figure 3B,C). These results were further confirmed by assessing the expression of RIG-I and MEX3A by immunohistochemistry staining of paratumor and GB tissue sections (Figure 3D,E). Accordingly, we observed a remarkable protein expression of MEX3A in three GB cell lines (U87, A-172 and T98G) compared to normal brain tissue (NBT) and normal brain cells (NBC) used as a control, which correlated with low RIG-I protein levels (Figure 3F,G).

Overall these data suggested MEX3A as a potential regulator of RIG-I protein.

### 2.3. MEX3A Impairs RIG-I Protein Stability

To investigate if MEX3A is involved in the regulation of RIG-I, we evaluated the effect of its ectopic expression on RIG-I protein levels. As shown in Figure 4A,B, increasing amounts of MEX3A lead to a robust reduction of RIG-I protein levels. Moreover, we compared the effects of MEX3A versus MEX3C and RNF122, and both are found to be up-regulated in GB (Appendix A and Figure 2B,C) and are known to mediate RIG-I stabilization and degradation, respectively [28,34].

Interestingly, MEX3A showed a robust effect on the reduction of RIG-I protein expression (Figure 4A,B), without affecting its mRNA levels (Appendix A), and it was able to counteract the effect of MEX3C on the stability of RIG-I when the two E3-ligases were co-expressed (Figure 4C,D).

We further confirmed the effect of MEX3A on RIG-I protein stability by cycloheximide (CHX) assay. Half-life of RIG-I was significantly decreased in presence of ectopic expression of MEX3A, suggesting the role of this E3 ligase in the degradation of RIG-I (Figure 4E,F). Accordingly, we observed an increase of endogenous RIG-I protein levels, following the genetic depletion of MEX3A in A-172 and T98G GB cell lines (Figure 4G,H). A significant increase of the endogenous RIG-I half-life was also observed in T98G cells genetically silenced for MEX3A after CHX treatment (Figure 4I,J). In all these conditions, no modulation of *RIG-I* mRNA levels was observed (Appendix A). Together, these findings indicated that MEX3A regulates RIG-I protein stability.

### 2.4. MEX3A Binds and Ubiquitylates RIG-I

To elucidate the molecular mechanisms through which MEX3A controls RIG-I stability, we tested the ability of MEX3A to interact with RIG-I. To verify this hypothesis, we carried out reciprocal co-immunoprecipitation (Co-IP) experiments in HEK293T cells transfected with Myc-MEX3A and GFP-RIG-I expression plasmids. The immunoblot analysis revealed that MEX3A binds RIG-I (Figure 5A,B). The interaction between endogenous MEX3A and RIG-I was also confirmed in A-172 and T98G GB cell lines (Figure 5C,D) and in GB tissues (Figure 5E).

Next, we verified whether MEX3A controls RIG-I protein levels by inducing its ubiquitin-dependent degradation. To this aim, we performed an in vivo ubiquitylation assay upon ectopic expression of MEX3A. Increasing amounts of MEX3A promoted the poly-ubiquitylation of RIG-I and its subsequent degradation by the proteasome, as indicated by the accumulation of RIG-I ubiquitylated forms and its protein levels in cells treated with the proteasome-inhibitor MG132 (Figure 5F,G).

Overall, these results indicated that MEX3A interacts with RIG-I and mediates its ubiquitylation, thereby inducing RIG-I proteasome-dependent degradation.

### 2.5. MEX3A Affects GB Cell Proliferation

To investigate the biological relevance of MEX3A function in GB, we analyzed the effect of its genetic depletion on the proliferation ability of two GB cell lines. We observed a decrease in cell growth over time and a significant reduction of BrdU uptake in A-172 cells silenced for MEX3A (shMEX3A) compared to the control cells (shCTR) (Figure 6A–D).

This was consistent with increased levels of the cleaved PARP protein (Figure 6E,F). Of note, also the migration ability of A-172 cells was significantly impaired following genetic depletion of MEX3A (Figure 6G,H). Importantly, similar results were observed in A-172 cells treated with cell cycle thymidine, demonstrating that this effect was independent of the decrease in cell proliferation due to MEX3A gene silencing (Appendix A). These data were confirmed also in T98G cell line (Figure 6I–P and Appendix A).

These data suggest the inhibition of MEX3A as a potential therapeutic approach for the treatment of GB.

## 3. Discussion

GB is the most common malignant primary brain tumor, driven by complex signaling pathways and particularly difficult to treat [4,35]. Although the current therapeutic protocols have been refined, GB remains a fatal human cancer, in which the estimated survival is 8 to 15 months [2,35]. Chemotherapy with Temozolomide (TMZ), the most used drug in GB treatment, shows acquired resistance caused by the high levels of activity of DNA repair enzyme O6-methylguanine DNA methyltransferase (MGMT) in tumor cells that reduces the effect of this alkylating agent. Unfortunately, other mechanisms contribute to the resistance to TMZ, such as overexpression of epidermal growth factor receptor (EGFR), galectin-1, MDM2, p53 and PTEN mutations [36]. Moreover, the significant heterogeneity of this tumor represents a potential source of therapeutic resistance and one of the main hindrances in the identification of new pharmacological targets for the development of more effective drugs. Furthermore, recent studies highlighted a complex crosstalk between tumor cells and the microenvironment, especially enhanced angiogenesis and aberration in anticancer immune response, which could be the cause of more aggressive tumor phenotype and failure of treatments effectiveness [2,5]. All of these issues represent a dramatic challenge for the development of new therapies and raise the need to improve the knowledge of the molecular mechanisms underlying GB tumorigenesis.

At present, it is clearly demonstrated that post-translational modifications play a crucial role in the control of normal cellular processes, but also in cancer initiation and progression [37,38,39,40]. 

Here, we described for the first time that the RNA-binding MEX3A affects GB proliferation, acting as an E3-ubiquitin ligase. So far, little is known about the role of this protein in carcinogenesis [21,22,23,24,26]. Although MEX3A is a member of the RNA-binding ubiquitin ligase MEX3 protein family, its function as E3-ligase and its potential substrates have not yet been described.

We found that MEX3A is strongly up-regulated in GB specimens compared to the normal brain tissues and regulates the protein stability of RIG-I. The multifunctional protein RIG-I is an important PRR involved in the activation of antiviral innate immunity and interferon response pathway [29], and known to play many other biological roles. Interestingly, recent studies have described that RIG-I activation by RNA ligands is essential in the induction of cell growth arrest via apoptosis and immune activation in several types of cancers, including GB [32,33,41]. 

In particular, Glas and colleagues [32,33,41] suggested that the activation of the innate immune system could occur by stimulation of innate immune receptors involved in antiviral and antitumor responses, such as RIG-I and MDA5. More intriguingly, they showed that the increased expression levels of RIG-I and MDA5 by interferon stimulation could mediate immune response and allow hitting different GB malignant cells, including cancer stem cells, with low level of neurotoxicity. Targeting the cytosolic innate immune receptors in GB stands as innovative strategy to overcome therapeutic resistance due to cellular heterogeneity and immune escape mechanisms [42,43].

RIG-I activity is significantly modulated through ubiquitylation processes mediated by several E3 ligases [34,44,45,46,47], differentially expressed depending on the tumor context [48,49]. MEX3C, TRIM4 and TRIM25 are the most important E3 ligases for positive regulation of RIG-I activity in response to viral infection, whereas RNF122 and RNF125 have been described to mediate the ubiquitin-dependent degradation of this cytosolic innate immune receptor. 

In our study, we found higher expression levels of MEX3A and RNF122, compared to other E3 ligases, and lower baseline protein levels of RIG-I in GB specimens than in paratumor tissues. This evidence suggests the prevalence of degradative rather than regulative processes on the control of RIG-I in GB.

We demonstrated that MEX3A interacts with ubiquitylates RIG-I, promoting its proteasome-dependent degradation and altering its stability. We showed that the gene silencing of MEX3A increases protein levels of RIG-I and strongly inhibits the proliferation of human GB cells (Figure 7).

In this scenario, the stimulation of RIG-I, as well as the design of MEX3A inhibitors or small molecules able to counteract the binding between MEX3A and RIG-I might represent a novel strategy for cancer immunotherapy, by improving the dual function of RIG-I as cell death inducer and immune response activator (Figure 7). Targeting RIG-I and MEX3A could open innovative perspectives for new multi-targeting approaches in the treatment of GB.

## 4. Material and Methods

### 4.1. Cell Cultures, Transfections and Lentiviral Infections

HEK293T (cod. CTR-3216^TM^) and human glioblastoma A-172 cells (purchased from the American Type Culture Condition, ATCC) were cultured in Dulbecco’s Modified Eagle Medium (DMEM, Sigma Aldrich (St. Louis, MO, USA) supplemented with 10% fetal bovine serum (FBS; Sigma-Aldrich). Human glioblastoma multiforme T98G and human glioblastoma-astrocytoma U-87, (purchased from ATCC) cells were maintained in Eagle’s minimum essential medium (MEM, Sigma Aldrich plus 10% FBS. All media contained 1% Penicillin–Streptomycin and 1% Glutamine (Invitrogen, Gaithersburg, MD, USA).

GB, paratumoral biopsies and their related information were obtained under written informed consent and ethical approved by the Internal Review Board of the Umberto I, Policlinico of Rome (#3623). Mycoplasma contamination in cell cultures was routinely monitored by using a PCR detection kit (Applied Biological Materials, Richmond, BC, Canada).

Transient transfections were performed by using DreamFect^TM^ Gold transfection reagent (Oz Biosciences SAS, Marseille, France) in accordance with the manufacturer’s protocol. 

Lentiviral particles were generated in HEK293 cells transfected with packaging and envelope plasmids (pCMV-dR8.74 and VSV-G/pMDG2), pGFP-pLKO.1 plasmids (shCTR TR30021; shMEX3A TL308061B (#1), TL308061C (#2), Origene, Rockville, MD, USA). A-172 and T98G cells were infected with complete medium of HEK293 cells containing lentiviral particles and polybrene (8 µg/mL, Sigma-Aldrich) for 72 h.

### 4.2. Plasmids, Antibodies and Treatments

The pCMV6-Entry-Myc-DDK-RNF122 (RC210868), pCMV6-Entry-Myc-DDK-MEX3A (RC215359), pCMV6-Entry-Myc-DDK-MEX3C (RC221125), pCMV6-AC-GFP-DDX58 (RG217615), non-effective Scrambled shRNA Cassette in pGFP-C-shLenti Vector (shCTR, TR30021), pGFP-C-shLenti-MEX3A (TL308061B (#1) and TL308061C (#2)) were purchased from Origene (Rockville, MD, USA).

Mouse anti-RIG-I D-12 (sc-376845, 1:1000 for WB, 1:100 for IHC), anti-RIG-I HRP conjugated D-12 (sc-376845, 1:2000), mouse anti-HA-probe F-7 HRP (sc-7392 HRP, 1:1000) and mouse anti-β-Actin C4 (sc-47778, 1:2000) were purchased from Santa Cruz Biotechnology (Santa Cruz, CA, USA). Anti-Flag M2 HRP (A8592, 1:1000) was purchased from Sigma-Aldrich. Rabbit anti-RIG-I (D14G6, 1:1000) and Rabbit anti-PARP (95426S) were purchased from Cell Signaling (Beverly, MA, USA). Rabbit anti-MEX3A (ab79046, 1:1000 for WB, 1:100 for IHC) was purchased from Abcam (Cambridge, UK). HRP-conjugated secondary antibodies were purchased from Bethyl Laboratories (Montgomery, TX, USA).

Where indicated, cells were treated with MG132 (50 µM; Calbiochem, Nottingham, UK) for 4 h, Cycloheximide (100 µM, Sigma Aldrich) at indicated time.

### 4.3. Immunoblot Analysis and Immunoprecipitation

For Immunoblot analysis, cells were lysed in a solution containing RIPA buffer (50 mM Tris-HCl at pH 7.6, 150 mM NaCl, 0.5% sodium deoxycholic, 5 mM EDTA, 0.1% SDS, 100 mM NaF, 2 mM NaPPi and 1% NP-40) supplemented with protease and phosphatase inhibitors. The lysates were incubated on ice and then centrifuged at 13,000 g for 30 min, at 4 °C. After centrifugation, a determined amount of supernatant was resuspended in sample loading buffer, boiled for 5 min, resolved in SDS-PAGE and then subjected to immunoblot analysis. For Co-immunoprecipitation (Co-IP) assays, cell pellets were lysed with Triton Buffer (50 mM Tris-HCl pH 7.5, 250 mM sodium chloride, 50 mM sodium fluoride, 1 mM EDTA pH 8 and 0.1% Triton X-100), supplemented with protease and phosphatase inhibitors. An established amount of the whole-cell protein extracts was immunoprecipitated overnight at 4 °C, with rotation with specific primary antibodies (2 µg/mg of protein lysate) or IgG used as a control (2 µg/mg of protein lysate; Santa Cruz Biotechnology) and then incubated with Protein agarose beads (Santa Cruz Biotechnology) for 1 h, at 4 °C, with rotation. The immunoprecipitates were then washed five times with the lysis buffer described above, resuspended in sample loading buffer, boiled for 5 min, resolved in SDS-PAGE and then subjected to immunoblot analysis.

Clinical samples for immunoblot analysis or Co-IP assay were mechanically disrupted in RIPA buffer or Triton buffer, respectively, supplemented with protease and phosphatase inhibitors. The lysates were sonicated for 10′’ at 100 W amplitude in ice and then centrifuged at 13,000 g for 30 min at 4 °C. The lysates were assayed with the procedures described above.

### 4.4. In Vivo Ubiquitylation Assay

HEK293T were transfected with equal amount of GFP-RIG-I and HA-Ub and with Myc-MEX3A in 1:20, 1:10 and 1:5 ratio compared to GFP-RIG-I amount. After 24 h from transfection, cells were treated with MG132 (50 µM) or DMSO as control, for 4 h. Cells were lysed in a solution containing RIPA buffer, as described above. Lysates were subjected to immunoprecipitation with mouse anti-RIG-I (Santa Cruz Biotechnology), overnight, at 4 °C, with rotation. The immunoprecipitated proteins were then washed five times with the RIPA lysis buffer, resuspended in sample loading buffer, boiled for 5 min, resolved in SDS-PAGE and then subjected to immunoblot analysis. Polyubiquitylated forms were detected by using mouse anti-HA from Santa Cruz Biotechnology.

### 4.5. Cell Proliferation Assay

Trypan blue count was performed in GB cell lines, silenced by infection with lentiviral particles encoding either control shRNA (shCTR) or MEX3A shRNAs (shMEX3A#1 and shMEX3A#2) to assess the growth rate and cell viability. BrdU incorporation assay (Roche, Welwyn Garden City, UK) was used to determine cell proliferation in A-172 and T98G cells. First, cells were infected with lentiviral particles encoding either short hairpin RNA targeting human MEX3A or a control non-targeting sequence (shCTR) for 72 h, pulsed 24 h with BrdU and then fixed and permeabilized with 4% paraformaldehyde and 0.2% Triton X-100 (Sigma Aldrich). Dako Fluorescent mounting (Dako, Carpinteria, CA, USA) was used as mounting medium, and nuclei were counterstained with Hoechst reagent; BrdU detection was performed according to the manufacturer’s instructions. At least 500 nuclei were counted in triplicate, and the number of BrdU-positive nuclei was recorded.

For IncuCyte^®^ S3 experiments, infected A-172 and T98G cells were seeded in 96-well plates (20 × 10^3^ cells/well for both cell lines; 12 wells for each condition) in complete medium and incubated overnight. The day after seeding, T98G cells were treated with IncuCyte^®^ NucLight Rapid Red Reagent (#4717, Essen BioScience, 1:1000). Plates were transferred into the IncuCyte^®^ S3 Live Cell Analysis Systems and incubated at physiological condition (37 °C, 5% CO_2_), over 72 h. Images were collected every 2 h, and proliferation was evaluated as number of NucLight positive cells for T98G, or as cell confluence for A-172, which does not incorporate NucLight reagent. The experiments were performed in triplicate, and data were analyzed by using the IncuCyte software package (Essen BioScience, Ann Arbor, MI, USA).

### 4.6. mRNA Expression Analysis

Total RNA was isolated from GB and paratumoral biopsies, using TRIzol reagent (Invitrogen/Life Technologies, Carlsbad, CA, USA), and reverse-transcribed with a SensiFAST cDNA Synthesis Kit (Bioline Reagents Limited, London, UK). Quantitative real-time PCR (qRT-PCR) analysis of *MEX3A*, *RNF122*, *MEX3C* and *RIG-I* mRNA expression was performed by using the ViiA^TM^ 7 Real-Time PCR System (Life Technologies). Standard qPCR thermal cycler parameters were used to amplify a reaction mixture containing cDNA template, SensiFAST SYBR Lo-ROX Kit (Bioline Reagents Limited, London, UK) and primer probe. Then, mRNA quantification was performed by using SDS version 2.3 software, each sample was amplified in triplicate, and the average of the three threshold cycles was used to calculate the number of transcripts. Data were normalized with the endogenous housekeeping genes (*GAPDH* and *HPRT*) and expressed as the fold change respect to the control sample value.

The following primers were used to amplify the indicated genes:*hMEX3A For* 5′- ATCGTGGGCAGG CAAGGCT-3′*hMEX3A Rev* 5′- GCTGCTGAGATGATTT CCC-3′ *hRNF122 For* 5′- CCAACAAGTCCTGCTCGATG-3′*hRNF122 Rev* 5′- ACTGCGCAGGTCTGCCCATA-3′*hMEX3C For* 5′ –TGAACGGGGAGCAGGCG-3′*hMEX3C Rev* 5′- TGACTTGGACGGTGGTTTGA-3′*hRIG-I For* 5′- TGTGCTCCTACAGGTTGTGGA-3G*hRIG-I Rev* 5I- CACTGGGATCTGATTCGCAAA A-3n*hGAPDH For* 5′- TCCCATCACCATCTTCCAGG-3′*hGAPDH Rev* 5′- ATGAGTCCTTCCACGATACC-3′*hHPRT For* 5′- ATTATGCTGAGGATTTGGAAAGGG-3′*hHPRT Rev* 5′- GCCTCCCATCTCCTTCATCAC-3′

### 4.7. Immunohistochemistry

Tissues used for immunohistochemical staining were fixed in formalin and paraffin-embedded (FFPE). FFPE slides were incubated with monoclonal antibodies against RIG-I or MEX3A (1:100, overnight, at 4 °C). The day after, the slides were incubated for 20 min, with secondary antibodies coupled with peroxidase (Dako). Bound peroxidase was detected by diaminobenzidine (DAB) solution (ScyTek Laboratories, Logan, UT, USA) and EnVision FLEX Substrate buffer containing peroxide (Dako). Cell quantification was performed on collected sections, using the imaging software NIS-Elements BR 4.00.05 (Nikon Instruments Europe B.V., Florence, Italy). Images were captured by HistoFAXS software (TissueGnostics GmbH, Vienna, Austria), at 20× magnification.

### 4.8. In Vitro Wound Healing Assay

A-172 and T98G cells were infected with lentiviral particles encoding short hairpin RNA targeting MEX3A (shMEX3A) or a control non-targeting sequence (shCTR). Then, 20 × 10^3^ cells/well were seeded into a 96-well ImageLock tissue culture plate (Essen BioScience; 12 wells for each condition) and incubated at 37 °C with 5% CO_2_ incubator, for 36 h, until they reached 100% confluence. Wounds were made by the 96-well WoundMaker (Essen BioScience). The wounded cells were washed twice to remove the detached cells, and then incubated at 37 °C, in complete medium, with or without Thymidine 5 mg/mL (89270-5G, Sigma-Aldrich), to block the proliferation. Images of the wounds were automatically acquired within the incubator by IncuCyte zoom software (Essen BioScience). The wound image updates were taken at 2 h intervals for the duration of the experiment. Data were analyzed with respect to wound confluence and calculated by using the IncuCyte software package (Essen BioScience).

### 4.9. Statistical Analysis

Statistical analysis was performed by using the StatView 4.1 software (Abacus Concepts, Berkeley, CA, USA). For all experiments, *p*-values were determined by two-tailed Student’s *t*-test, and statistical significance was set at *p* < 0.05. Results are expressed as mean ± SD from an appropriate number of experiments (at least three biological replicas).

## 5. Conclusions

In this work, we identified new molecular players involved in GB tumorigenesis. The RNA-binding ubiquitin ligase MEX3A was found up-regulated in GB specimens, and this correlates with low protein levels of the tumor-suppressor RIG-I. We showed that MEX3A binds and ubiquitylates RIG-I, inducing its proteasome degradation. Genetic depletion of MEX3A results in the increase of RIG-I protein levels and the inhibition of GB cell growth. Our study unveils a new molecular mechanism involved in GB tumorigenesis and suggests that targeting RIG-I and MEX3A could pave the road for new therapeutic strategies for GB.

## Figures and Tables

**Figure 1 cancers-12-00321-f001:**
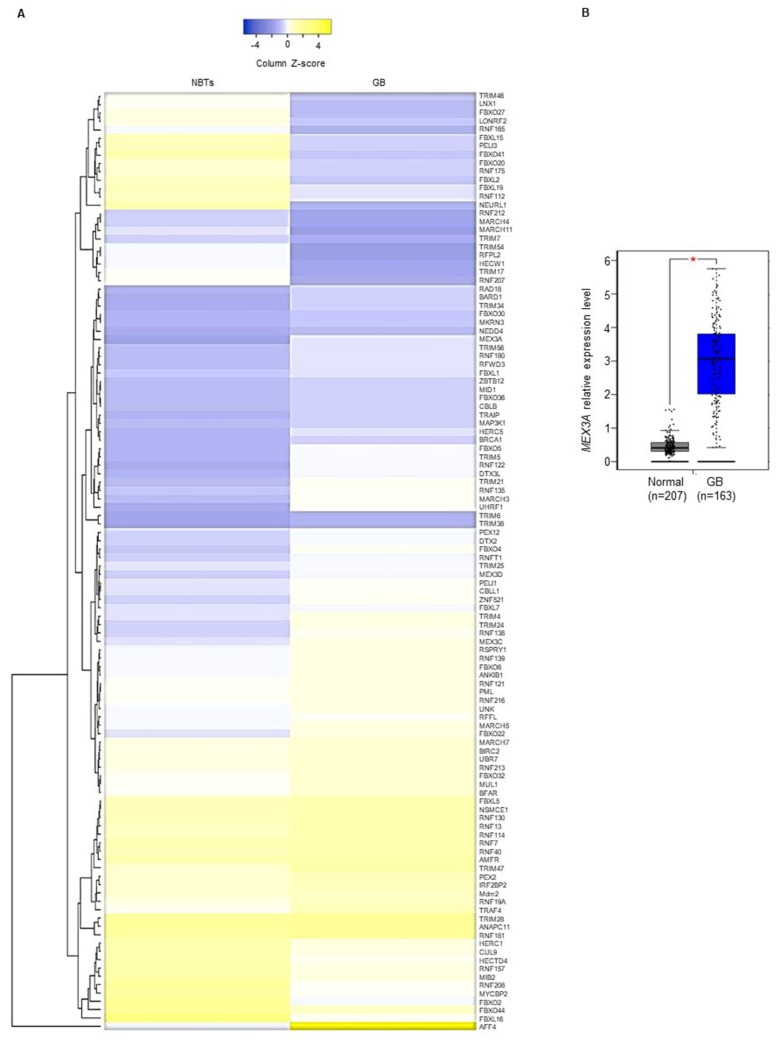
Gene expression profile of E3-ligases and F-box proteins in GB. (**A**) Heatmap shows a Z-score transformed gene expression values of catalytic E3-ligases and F-box proteins between normal brain tissues (NBTs) and GB specimens. Average linkage was used as hierarchical clustering method with Euclidean distance measurement. Color scale bar indicates the intensity associated with normalized expression values. (**B**) Gene expression of *MEX3A* in GB compared to NBTs. * *p* < 0.05. All the data were retrieved from Gene Expression Profiling Interactive Analysis (GEPIA) (http://gepia.cancer-pku.cn/).

**Figure 2 cancers-12-00321-f002:**
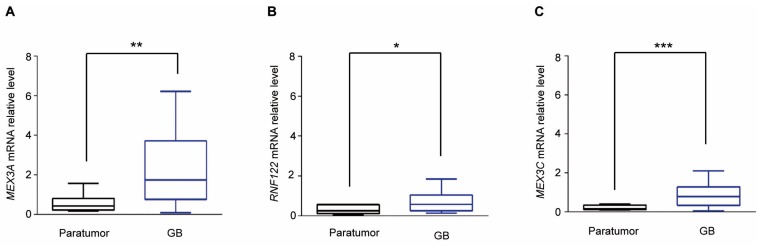
MEX3A expression is up-regulated in GB specimens. (**A**–**C**) Quantitative real-time PCR (qRT-PCR) analysis of *MEX3A*, *RNF122* and *MEX3C* gene expression in GB samples described in Table 1 compared to corresponding paratumor tissues. Data are normalized to endogenous *GAPDH* and *HPRT* controls. Mean ± SD; * *p* < 0.05; ** *p* < 0.01; *** *p* < 0.001 calculated with two-sided Student’s t-test.

**Figure 3 cancers-12-00321-f003:**
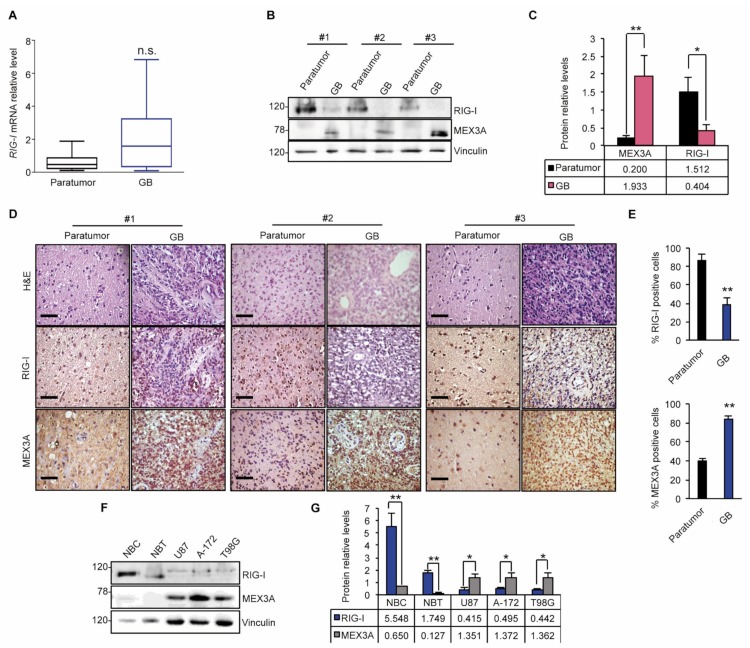
MEX3A up-regulation is associated with low RIG-I protein level in GB. (**A**) qRT-PCR gene expression analysis of *RIG-I* in GB specimens shown in Table 1. Data were normalized to endogenous *GAPDH* and *HPRT* controls. (**B**) Representative immunoblotting analysis of MEX3A and RIG-I proteins levels in three GB and paratumor paired samples. (**C**) Densitometric analysis of actin-normalized of MEX3A and RIG-I protein levels shown in (**B**). (**D**) H&E and immunohistochemical staining of MEX3A and RIG-I of the GB and paratumor samples analyzed in (**B**). Scale bar 100 μm. (**E**) Quantification of immunohistochemical staining shown in (**D**). (**F**) MEX3A and RIG-I steady state in normal brain tissue (NBT) and normal brain cells (NBC) compared to GB cell lines. (**G**) Densitometric analysis of actin-normalized protein levels assayed in (**F**) represents the mean of three independent experiments. Mean ± SD; * *p* < 0.05; ** *p* < 0.01 determined with two-sided Student’s t-test.

**Figure 4 cancers-12-00321-f004:**
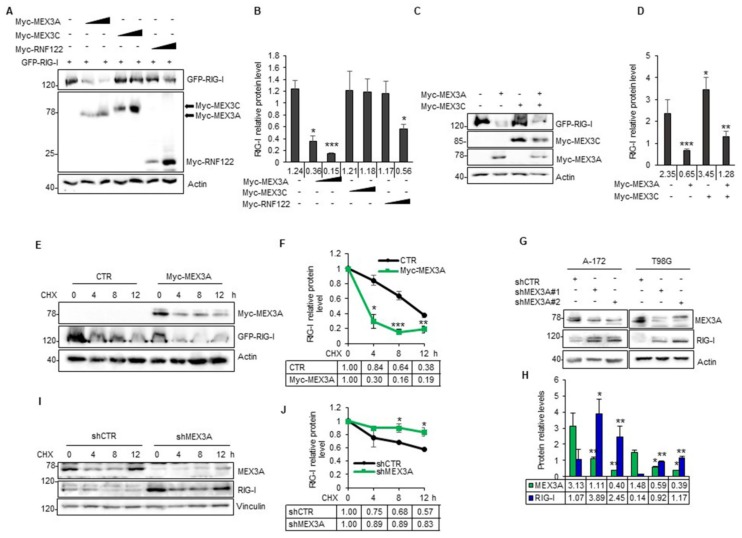
MEX3A expression affects RIG-I protein stability. (**A**,**B**) Representative immunoblotting (**A**) and densitometric analysis (**B**) of GFP-RIG-I protein level in HEK293T cells transfected with increasing amounts of Myc-tagged MEX3A, MEX3C and RNF122, in a (1:1) and (1:0.5) ratio, compared to the amount of the transfected GFP-RIG-I. (**C**) HEK293T cells were transfected with equal quantities of GFP-RIG-I and Myc-tagged MEX3A and MEX3C, alone or in combination. The levels of the indicated protein were assessed by immunoblot analysis. (**D**) Densitometric analysis of GFP-RIG-I protein levels shown in (**C**) and normalized on actin levels. (**E**) HEK293T cells were co-transfected with GFP-RIG-I, an empty vector as control or Myc-MEX3A, in a ratio (1:0.5). Twelve hours after transfection, GFP-RIG-I half-life was assayed, following treatment with CHX (100 µg/mL), at the indicated time. (**F**) Densitometric analysis of RIG-I protein levels shown in (**E**) and normalized on actin levels. (**G**) Protein expression levels of endogenous MEX3A and RIG-I in A-172 and T98G GB cell lines transduced with lentiviral vectors encoding either control shRNA (shCTR) or MEX3A shRNAs (shMEX3A#1 and shMEX3A#2). (**H**) Densitometric analysis of RIG-I and MEX3A protein levels shown in (**G**) and normalized on actin levels. (**I**) Half-life of endogenous RIG-I in T98G cells infected with lentiviral particles encoding shCTR or shMEX3A#2. Cells were treated with CHX (50 µg/mL) at the indicated time points. (**J**) Densitometric analysis of the RIG-I protein levels shown in (**I**). All the densitometric analysis represent the mean of protein levels of three independent experiments. Mean ± SD; * *p* < 0.05; ** *p* < 0.01; *** *p* < 0.001 calculated with two-sided Student’s t-test.

**Figure 5 cancers-12-00321-f005:**
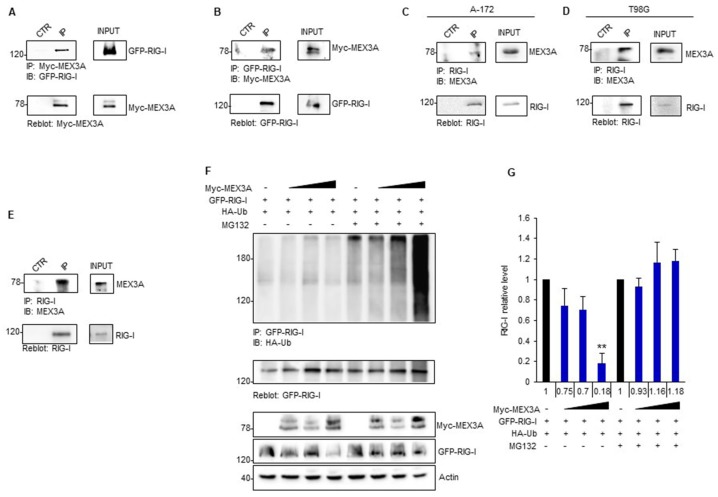
MEX3A interacts with RIG-I mediating its ubiquitylation. (**A**,**B**) HEK293T cells were co-transfected with GFP-RIG-I and Myc-MEX3A in a (1:0.5) ratio, respectively. Interaction between RIG-I and MEX3A was detected by immunoprecipitation, followed by immunoblot analysis with the indicated antibodies. (**C**,**D**) Interaction between endogenous RIG-I and MEX3A was assessed by immunoprecipitation and immunoblotting in A-172 (**C**) and T98G (**D**) GB cell lines. (**E**) Binding between endogenous RIG-I and MEX3A was assessed by co-immunoprecipitation assay in GB clinical sample. (**F**) GFP-RIG-I was immunoprecipitated from HEK293T expressing the indicated proteins and treated with MG132 (50 μM) or control for 4 h, followed by immunoblotting with an anti-HA antibody to detect conjugated HA-Ub. Ubiquitylation blot was re-probed with a GFP antibody, to detect the immunoprecipitated level of RIG-I. Bottom GFP-RIG-I and Myc-MEX3A protein levels in total cell lysate. (**G**) Densitometric analysis of the levels of the indicated protein shown in (**F**) represents the mean of three independent experiments. Mean ± SD; ** *p* < 0.01 calculated using two-tailed Student’s *t*-test.

**Figure 6 cancers-12-00321-f006:**
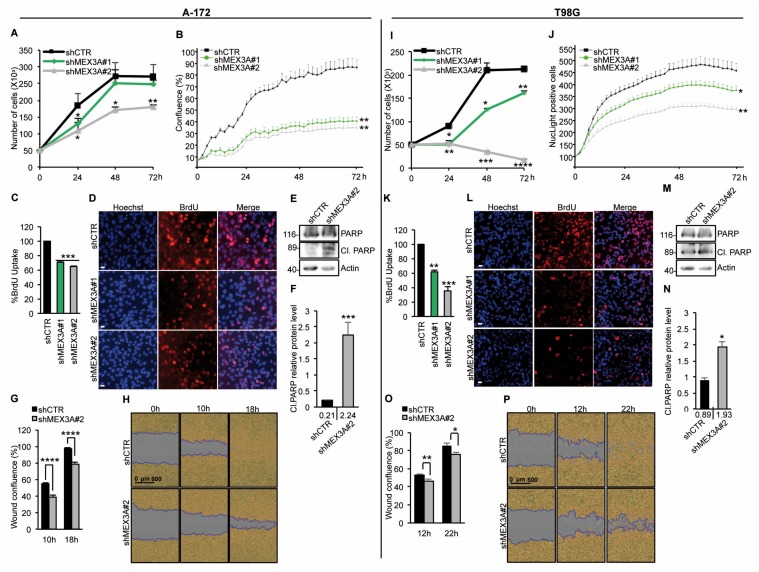
MEX3A impinges the proliferation of human A-172 and T98G GB cells. (**A**,**I**) A-172 (**A**) and T98G (**I**) cells were counted by trypan blue at the indicated time points to evaluate the growth rate after infection with lentiviral particles encoding either control shRNA (shCTR) or MEX3A shRNAs (shMEX3A#1 and shMEX3A#2). (**B**,**J**) Infected A-172 (**B**) and T98G (**J**) cells were seeded in 96-well tissue culture plates, and their proliferation was measured as cell confluence (%) calculated using IncuCyte Zoom software by phase-contrast images. Cells were scanned every two hours from 0 to 72 h after infection. The graph shows data ± SD analyzed by using the IncuCyte software (Essen BioScience). (**C**,**K**) The graphs show the percentage of BrdU uptake in A-172 (**C**) and T98G (**K**) transduced with lentiviral particles encoding either control shRNA (shCTR) or MEX3A shRNAs (shMEX3A#1 and shMEX3A#2). (**D**,**L**) Representative immunofluorescence images of the BrdU incorporation shown in (**C**,**K**), respectively. Scale bar: 5 µm. (**E**,**F** and **M**,**N**) Immunoblot and densitometric analysis of cleaved PARP protein in A-172 (**E**,**F**) and T98G (**M**,**N**) cells transduced with lentiviral vector encoding either control shRNA (shCTR) or MEX3A shRNA#2. (**G**,**O**) Infected A-172 (**G**) and T98G (**O**) cells were seeded in 96-well tissue culture plates and cultured for 36 h, to reach 100% confluence. Then, cells were scratched, and images of wounds were automatically acquired within the CO2 incubator by IncuCyte zoom software. The wound image updates were taken at 2 h intervals for the duration of the experiment. The graph shows data ± SD analyzed with respect to wound confluence and calculated by using the IncuCyte software package. (**H**,**P**) Representative images of the data are shown in (**G**,**O**), respectively. All data represent the mean of three independent experiments. Mean ± SD; * *p* < 0.05; ** *p* < 0.01; *** *p* < 0.001; **** *p* < 0.0001 calculated with two-sided Student’s *t*-test.

**Figure 7 cancers-12-00321-f007:**
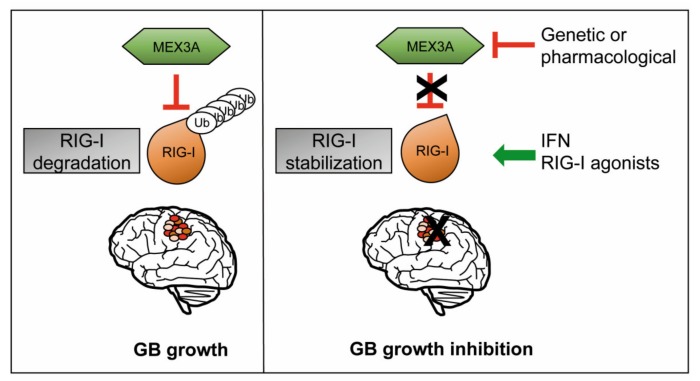
A representative model showing the role of MEX3A and RIG-I in GB tumorigenesis. RNA-binding ubiquitin ligase MEX3A ubiquitylates and induces the proteasome degradation of RIG-I, thus favoring GB growth. On the other hand, genetic depletion of MEX3A results in the increase of RIG-I protein levels impairing GB proliferation. The inhibition of MEX3A and/or the activation of RIG-I could represent promising multi-targeting strategies for GB treatment.

**Table 1 cancers-12-00321-t001:** Characteristics of 27 high-grade GB specimens.

Characteristics	Number of Patients (27)
**Age years**	(%)
<60	44.45
≥60	55.55
**Sex**	
Male	51.85
Female	48.15
**IDH1 status**	
Wild Type	77.77
Mutant	7.4
n.d.	14.83
**EGFR expression**	
Not expressed	22.22
Expressed	22.22
Highly expressed	25.92
n.d.	39.63
**p53 expression**	
Not expressed	37.03
Very Low Expressed	7.4
Expressed	40.74
n.d. *	14.83

* n.d. not determined.

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
