# Peer review of "The RNA-Binding Ubiquitin Ligase MEX3A Affects Glioblastoma Tumorigenesis by Inducing Ubiquitylation and Degradation of RIG-I"

_cancers, 2020, doi:10.3390/cancers12020321_

Round 1

Reviewer 1 Report

The paper of Bufalieri et al. identifies MEX3A/RIG-I complex as a novel molecular mechanism involved in glioblastoma and a promising therapeutic target in GBM.

The issue is extremely relevant, being GBM a deadly disease with a median overall survival of 14 months.

The paper is well written, and experiments clearly described.

Some minor concerns:

Minor misspellings are presents (e.g. an tumor suppressor; and in immune activation; growth rate and cell viability.BrdU incorporation assay; analysis of MEX3A, RNF122, MEX3C and RIG-I mRNA; Data were normalized with the endogenous housekeeping genes (GAPDH and HPRT)) Statistics: authors used parametric tests. Did they check for normality before applying parametric tests? I would change the title of Figure 3 legend, “MEX3A up-regulation is related to low RIG-I protein level in GBM”. I think that “MEX3A up-regulation is associated with low RIG-I protein level in GBM” is more appropriated for the data so far shown. Figure 3C: from the histograms it is not apparent that “Data was expressed as the fold change respect to the paratumor samples”. Figure 3F-G: I think that NBT is not the best control for cell lines. I would suggest the use of non-tumor cell lines. Figure 3G: the symbol “****” on tumor cell lines indicate significant difference between RIG1 and MEX3A in the cell lines, or between NBT and cell lines, for both RIG1 and MEX3A? “..an inverse correlation was observed in the corresponding paratumor samples (Figure 3B and C)”: did you use a statistical test to support this statement? Otherwise I would use “trend” instead of “correlation”. Methods: authors cite MEFs cells, but it is not clear in which experiments they were used. Figure 6 and figure 7: check “Cl PURP” in the figure. From figure 6A and 6B, it looks like MEX3A is inducing a change in cell volume, rather than in cell number.

Reviewer 2 Report

Comment: 1291 words

[Cancers] Manuscript ID: cancers-690233, (authors correspondence: Paola Infante *, Lucia Di Marcotullio *), attempted to initiate their inquiry with 27 GBM specimens, on MEX3A/RIG-I complex as GBM biomarkers, as evidence that concomitant phenomena of high MEX3A and low RIG-I protein level were manifested in GBM specimens. With the ectopic expression system, they “demonstrate that MEX3A interacts and ubiquitylates RIG-I, promoting its proteasome-dependent degradation.” The pulldown assay showed MEX3A binds RIG-I and induced RIG-I ubiquitylation and proteasome-dependent degradation. They further found that the gene silencing of MEX3A strongly inhibits the proliferation and migration of human GBM cells in vitro.

They stretched out to conclude that MEX3A/RIG-I complex as a promising therapeutic target. Many questions remain to be elucidated: how can they spatiotemporally target a complex? How do they speculate on the balance of such a complex?

It’s an innovative report. They articulated that “At present, ubiquitylation substrates of MEX3A have not been identified. We first hypothesized that this E3-ubiquitin ligase could be involved in the regulation of RIG-I, a tumor suppressor that controls differentiation, apoptosis, and tumorigenesis processes. This idea was supported by the evidence that RIG-I stability and activity is modulated by the E3-ligases RNF122 [34] and MEX3C (the MEX3A paralogue) [20], whose mRNA expression levels were found up-regulated in GBM (Supplementary Figure 1B and C and Figure 2B and C) [28,33,35,36].” Thus, it is an interesting report. However, they need to address the following 13 specifics to improve its clarity and coherence.

Specific comments:

MEX3A/RIG-I complex as a promising therapeutic target Introduction: “high resistance to standard treatments” – the authors should elaborate on how the resistance occurs with their insights. For example, how does RIG-I, acting as a tumor suppressor involved in differentiation, apoptosis, and innate immune response, play what role in such resistance? What’s the cause or consequence (Refer to Fig. 3)? Retinoic acid-inducible gene I (RIG-I) as the unique substrate of RNA-binding ubiquitin ligase MEX3A, the ubiquitylation of RIG-I mediated by MEX3C leads to RIG-I activation without interfering with its degradation and protein stability. How does this feedback regulation work in normal and abnormal (GBM) differentially so promising in therapeutic targeting? “RIG-I is a critical cytosolic pattern recognition receptor (PRR) that acts as an RNA sensor to activate innate antiviral immunity and interferon (IFN) production [29].” It sounds a spatiotemporal mechanism – an illustrative scheme should be used to show the stepwise feedback and links possible for targeted drugs. Can they speculate on the “depletion of MEX3A results in the impairment of GBM cell proliferation?” Their scheme should be included. Table 1, “Table 1. Characteristics of 27 high-grade GBM specimens.” Can they give out the concomitant expression of IDH1, EGFR, and p53 in those GBM specimens? Any drug-driven mutations? Fig. 3 and related data didn’t adequately define that “Overall, these data strongly suggested MEX3A as a good candidate for the negative post-translational regulation of RIG-I.” these are more circumstantial, not consequential data sets. “As shown in Figures 4A and 4B, increasing amounts of MEX3A lead to a robust reduction of RIG-I protein levels.” Their systems (HEK293T cells transfected with increasing amounts of Myc-tagged MEX3A, MEX3C, and RNF122. in A-172 and T98G GBM cell lines transduced with lentiviral vectors) did not exclude squelching effects of artificial systems. Neither could they conclude only on cycloheximide (CHX) assay without mRNA data. “Of note, also the migration ability of A-172 cells was significantly impaired following genetic depletion of MEX3A (Figure 6G and H). Similar results were obtained in the T98G cell line (Figure 7).” How can they explain MEX3A affect proliferation and migration spatiotemporally in vivo? What are their Mechanistical studies? As noted, “Furthermore, recent studies highlighted complex crosstalk between tumor cells and the microenvironment, especially enhanced angiogenesis and aberration in anticancer immune response, which could be the cause of more aggressive tumor phenotype and failure of treatments effectiveness [2,5].” How did they weigh in on TME-mediated MEX3A-RIG-I complex? Given the fact “RIG-I is a multifunctional protein, which beyond being an important PRR, involved in the activation of antiviral innate immunity and interferon response pathway [29], it plays many other biological roles,” how could they target such a target? “In conclusion, our study unveils the involvement of MEX3A/RIG-I interaction in GBM tumorigenesis and suggests that targeting this complex could open innovative perspectives for new multi-targeting approaches in the treatment of GBM.” This is an overstatement, not supported by their experimental data. They need to provide in vivo data to show “MEX3A/RIG-I interaction in GBM tumorigenesis” – as their current data unable to establish “GBM tumorigenesis” but only speculation. They should focus their discussion on their data to expand on MEX3A/RIG-I interaction in GBM tumorigenesis and progression, ideally with an illustrative diagram, so that they get tighten up on tumorigenesis or nothing is related to that “MEX3A/RIG-I interaction in GBM tumorigenesis” – where is their data on GBM tumorigenesis – only in vivo data can establish such evidence. Some grammatical errors crawled on the text, English language and style are fine/minor spell check required. For example, “We first hypothesized that this E3-ubiquitin ligase could be involved in the regulation of RIG-I, an tumor suppressor that controls differentiation, apoptosis, and tumorigenesis processes.” [The use of articles was wrong: it is a, not an]. They need to follow the commonly-used term, such as “gene silencing,” not “genetic silencing.” Spelling: not cyclohexamide (CHX), but cycloheximide (CHX).

Reviewer 3 Report

In this manuscript the authors perform a gene expression analysis of ubiquitin ligases using publicly available datasets and identify the RNA-binding ubiquitin ligase, MEX3A (among others) as being over-expressed in glioblastoma (GB). Using a number of analysis and methodological approaches the authors show that MEX3A regulates the ubiquitination and subsequent proteasome-dependent degradation of RIG-1 protein, a tumor suppressor involved in differentiation, apoptosis and innate immune response. They further show that enhanced expression of RIG-I protein as a result of MEX3A depletion results in the suppression of GBM cell growth. Overall the results are robust and the authors use many approaches and methodologies to show the regulation of RIG-1 by MEX3A protein. The authors´ findings are novel and very interesting, they show the molecular mechanism for the regulation of RIG-I by MEX3A and a functional effect in GBM cells (regulation of cell growth). I feel that this manuscript is appropriate for publication in Cancers journal pending revisions.

Note: There is no line counting in the text.

Introduction:

Where it reads: “However, these approaches show very limited effectiveness, due to high rate of relapse, subsequent injuries of the patient’s neurological and physiological status”, it should read: However, these approaches show very limited effectiveness, resulting in a high rate of relapse and subsequent deterioration of the patient’s neurological and physiological status

Where it reads: “Deregulation or mutations of E3-ubiquitin ligases have been associated to several human tumors; for these reasons”, it should read:  Deregulation or mutations in E3-ubiquitin ligases have been associated with several human tumors; for this reason,

Where it reads: “At present, few information is available about”, it should read: At present, very little information is available about

Where it reads: “Further, RIG-I works as a tumor suppressor in several tumor types”, it should read: Furthermore, RIG-I works as a tumor suppressor in several tumor types

Results

Table 1: Where it reads: “No expressed” it should read: Not expressed or alternatively no expression

In Figure 3B the MEX3A western blots do not have publication quality, these blots need to be improved, particularly since this is the main protein in the manuscript.

Figure 3G legend the authors fail to state how many times the experiment in 3F was performed for quantification, this information should be added.

In Figure 4F, the authors should calculate and indicate the half-life of RIG-I in the absence (endogenous levels) or presence (over-expression) of MEX3A

In Figure 4G, the quality of the immunoblots for RIG-1 and MEX3A should be improved, as this is a fairly straightforward experiment and the quality of the current immunoblots is poor for publication.

The authors state: “Interestingly, MEX3A showed a higher effect on the reduction of RIG-I protein expression than ectopic RNF122 (Figure 4A and 4B)”. The authors cannot draw this conclusion, as it is impossible to know if the over-expression of the two proteins (MEX3A and RNF122) is similar by western blotting, there are many variables such as antibody detection, protein transfer (the proteins have quite different molecular weights), different half-lives of these proteins etc that need to be taken into consideration. In this way the more modest effect of RNF122 could be due to lower expression of this protein in the cells compared to MEX3A. For this reason it is not appropriate to take the conclusion above, as such I would suggest removing this sentence.

The authors state: “Half-life of RIG-I was significantly decreased in presence of ectopic expression of MEX3A, suggesting the role of this E3 ligase in the degradation of RIG-I (Figure 4E and F).” However they did not calculate the half-life of RIG-I in the two different situations, this should be added to the manuscript.

In Figure 5 legend the authors fail to explain what is their control for the co-immunoprecipitation experiments, this should be detailed in the figure legend. They should also show the IgG blots in order to demonstrate that similar amounts of antibody were added in the control compared to immunoprecipitation lanes. This is an important control. Suggestion: the authors could also add a higher expression of MEX3A (at least in one of the experiments) that lead to higher reduction of RIG-I to show a dose dependent effect. One would expect lower co-immunoprecipitation due to RIG-I degradation.

In Figure 5A the blot for RIG-I input lane is very strong. Taking into account the molecular weight of this protein (~20 KDa) the authors should check if this is correct, or if it is the IgG low molecular weight band that has been selected by mistake.

In Figure 5 legend: “(E) Binding between endogenous RIG-I and MEX3A was assessed by co-immunoprecipitation assay in GBM sample.” Is this a clinical sample? How was this done? Could you please clarify the procedure?

I would like to suggest merging Figures 6 and 7 together, since they show the exact same experiments using two different GB cell lines. It would make more sense to show this data side to side, instead of putting them separately.

The authors state: “Of note, also the migration ability of A-172 cells was significantly impaired following genetic depletion of MEX3A (Figure 6G and H). “ In order to say this the authors need to pre-treat the cells with a cell cycle inhibitor (for instance mitomycin-C), otherwise they cannot distinguish if the filling of the scratch was due to migration or simply proliferation into the empty area of the plate. Since, the authors show differences in cell proliferation, it is crucial to inhibit this before drawing any conclusions regarding cell migration.

General comments:

The authors use the nomenclature GBM to refer to glioblastoma, this is the old nomenclature referred to glioblastoma multiforme. The current nomenclature for glioblastoma is GB.

Overall the manuscript (including figure legends) requires editing of the English language.

In the Figure legends the authors do not indicate if the error bars are Standard Deviation or Standard error of the mean.

The transfection and infection protocols need more detail, as a general rule there should be enough detail so that other researchers could be able to reproduce the results. The authors do not indicate the amount of plasmids used in the experiments in the Figure legends or materials and methods sections. This is crucial for others to repeat these experiments. It is also important to know the amount of plasmids used in the over-expression experiments as compared to the co-immunoprecipitation experiments, as over-expression of MEX3A leads to down-regulation of RIG-I and as such it is crucial to understand the reproducibility and comparison between different experiments. In fact this will give more credibility and strength to these findings.

There are formatting issues in the materials and methods section of the manuscript.

The primers for qPCR analysis are missing in the materials and methods section.
